# Exploring The Relative Astringency of Tea Catechins and Distinct Astringent Sensation of Catechins and Flavonol Glycosides via an In Vitro Assay Composed of Artificial Oil Bodies

**DOI:** 10.3390/molecules27175679

**Published:** 2022-09-02

**Authors:** Chao-Tzu Liu, Jason T.C. Tzen

**Affiliations:** Graduate Institute of Biotechnology, National Chung-Hsing University, Taichung 402, Taiwan

**Keywords:** artificial oil bodies, astringency, caleosin, catechins, histatin 3

## Abstract

Artificial oil bodies covered by a recombinant surface protein, caleosin fused with histatin 3 (a major human salivary peptide), were employed to explore the relative astringency of eight tea catechins. The results showed that gallate-type catechins were more astringent than non-gallate-type catechins, with an astringency order of epicatechin gallate > epigallocatechin gallate > gallocatechin gallate > catechin gallate > epigallocatechin > epicatechin > gallocatechin > catechin. As expected, the extension of brewing time led to an increase in catechin content in the tea infusion, thus elevating tea astringency. Detailed analysis showed that the enhanced proportion of gallate-type catechins was significantly higher than that of non-gallate-type catechins, indicating that tea astringency was elevated exponentially, rather than proportionally, when brewing time was extended. Rough surfaces were observed on artificial oil bodies when they were complexed with epigallocatechin gallate (a catechin), while a smooth surface was observed on those complexed with rutin (a flavonol glycoside) under an atomic force microscope and a scanning electron microscope. The results indicate that catechins and flavonol glycosides induce the sensation of rough (puckering) and smooth (velvety) astringency in tea, respectively.

## 1. Introduction

Tea (*Camellia sinensis*), one of the most popular beverages around the world, mainly consists of caffeine, catechins and flavonol glycosides [1]. Oolong tea is a semi-fermented tea, in comparison with fully fermented black tea and non-fermented green tea [2]. The major constituents, catechins and other minor compounds in tea, have been shown to exhibit several biological functions, such as anti-cancer, anti-obesity, anti-oxidant, and anti-viral activities [3,4,5,6,7,8,9]. Eight catechins are commonly found in tea, including (−)-epicatechin (EC), (−)-epigallocatechin (EGC), (−)-epicatechin gallate (ECG), (−)-epigallocatechin gallate (EGCG), (−)-gallocatechin gallate (GCG), (−)-gallocatechin (GC),(−)-catechin gallate (CG) and (+)-catechin (C) [10]. These are divided into major and minor catechins, according to their contents in tea. Major catechins, including EGCG, EGC, ECG and EC, can be converted to minor catechins, such as GCG, GC, CG and C, via epimerization, which may occur during brewing processes, as well as in canned storage conditions [11]. Catechins are also divided into non-gallate-type catechins (EGC, EC, GC and C) and gallate-type catechins (EGCG, ECG, GCG and CG). Gallate-type catechins are presumably derived from galloylation of non-gallate-type catechins putatively catalyzed by epicatechin:1-O-galloyl-β-D-glucose O-galloyltransferase [12,13].

Astringency, which is part of oral tribology, is normally induced by phenolic compounds abundantly found in unripe fruits, red wine and tea [14,15,16]. Phenolic compounds form aggregates through interaction with human salivary proteins, and thus reduce the lubrication of saliva by blocking the mucosal pellicle on the oral epithelium [17,18,19,20]. The mechanism for the feeling of dryness and puckering in the mouth remains unknown [21,22]. However, some recent evidence showed that astringency is evoked by G-protein couple receptors and ion channels on taste buds and transported by the trigeminal nerve [23]. The sensation of puckering astringency is triggered by flavan-3-ols, such as catechins, whereas the sensation of velvety astringency is induced by flavonol glycosides [24]. It has been proposed that the galloyl groups on phenolic compounds are the major determinant for the distinct sensation between catechins and flavonol glycosides [25,26]. The sensation of rough (puckering) astringency may be activated by both trigeminal chemosensors and mechanoreceptors, whereas smooth (velvety) astringency might exclusively be activated through trigeminal mechanosensation [23,27]. A few methods were developed to evaluate astringency, such as a chemical reaction (gallic acid equivalence method) using the Folin–Ciocalteu reagent to quantitate phenolic compounds [28]; sensory evaluation, with the assistance of model solutions [29]; and an electronic tongue with different concentrations of tannic acid to define the standard curve of astringency [30]. Thus far, these oversimplified methods are not practically satisfactory in the determination of tea astringency.

Human saliva consists of electrolytes, enzymes, immunoglobulins and proteins [31]. Among salivary proteins, proline-rich proteins, as well as small peptides such as histatins and statherin, are able to interact with phenolic compounds, leading to the oral sensation of astringency [32]. The interaction between polyphenols and peptides can be divided into three stages [33]. When several polyphenol molecules bind to the same type of peptide, two peptides tend to crosslink due to additional polyphenols, acting as a link between peptides through weak intermolecular binding interactions. Therefore, the protein–polyphenol aggregates form spontaneously. Histatins of approximately 3 kDa with anti-bacterial and anti-fungal activities are secreted by parotid, submandibular and sublingual glands [34]. Recently, an in vitro assay was developed to estimate the relative astringency of tea infusions via artificial oil bodes that constituted of sesame oil sheltered by caleosin (an integral seed oil body protein) fused with histatin 3 [35]. The relative astringency of tea was measured by detecting the flotation of aggregated artificial oil bodies, which formed a visible emulsion layer on top of the solution, with its thickness in proportion to tea astringency. In this study, the in vitro assay was employed to evaluate the relative astringency of eight catechins in tea. The effect of an extended brewing time on tea astringency was also examined. Furthermore, surface roughness on artificial oil bodies when they were complexed with either epigallocatechin gallate (a catechin) or rutin (quercetin-3-*O*-rutinoside, a flavonol glycoside) was observed under an atomic force microscope and a scanning electron microscope, to explore the distinct sensation of rough and smooth astringency.

## 2. Results

### 2.1. Detection of Relative Astringency of Eight Catechins in Tea

Previously, a recombinant caleosin–histatin 3 fusion protein was engineered and used to generate artificial oil bodies for the detection of relative astringency of tea infusion [35]. In this study, artificial oil bodies were employed to evaluate the relative astringency of eight catechins commonly found in tea. The flotation of aggregated artificial oil bodies led to the formation of a visible emulsion layer (Figure 1A). The aggregation of artificial oil bodies interacting with each catechin was examined under a light microscope (Figure 1B). The relative astringency, calculated by measuring the thickness of emulsion layer in each catechin interaction, was detected and compared with ECG interaction regarded as 100% (Figure 1C). The results indicated that gallate-type catechins were more astringent than non-gallate-type catechins, with the astringency order of epicatechin gallate > epigallocatechin gallate > gallocatechin gallate > catechin gallate > epigallocatechin > epicatechin > gallocatechin > catechin.

### 2.2. Relative Astringency of Tea Infusion under Elongated Brewing Time

It is commonly known that extension of brewing time results in an increase in catechin contents in tea infusion, and thus elevates tea astringency. To assess the effect of brewing time on tea astringency, relative astringency was measured by preparing tea infusion from the same tea with brewing times of 10, 20, 30 and 40 min. The results showed that the emulsion layer on the upper portion of the cuvette gradually became thicker and thicker along with the increase in brewing time (Figure 2A). More and more aggregation of artificial oil bodies was observed when the brewing time was subsequently increased (Figure 2B). The thickness of the emulsion layer, representing relative astringency in each tea infusion was measured and compared (Figure 2C). The data showed that the relative astringency increased progressively as the brewing time was extended.

As expected, the contents of the four major catechins (EGCG, EGC, ECG and EC) increased progressively when the brewing time was extended, as revealed in the high-performance liquid chromatography (HPLC) analysis (Figure 3A). Detailed analysis showed that the enhanced proportion (107% and 145%) of gallate-type catechins (EGCG and ECG) was significantly higher (77% and 65%) than that of non-gallate-type catechins (EGC and EC) when the brewing time was increased from 10 min to 40 min, indicating that tea astringency was elevated exponentially instead of proportionally when brewing time was extended (Figure 3B).

### 2.3. Surface Roughness of Artificial Oil Bodies in Complex with EGCG or Rutin

Surface roughness of artificial oil bodies alone or complexed with EGCG (a catechin) or rutin (a flavonol glycoside) was observed under an atomic force microscope. A rough surface was observed on artificial oil bodies (Figure 4A). Similarly, a rough surface was also observed on artificial oil bodies when they were complexed with EGCG (Figure 4B). In contrast, a smooth surface was observed on artificial oil bodies complexed with rutin (Figure 4C). Quantitative calculation indicated that surface roughness of artificial oil bodies interacting with rutin was significantly lower than that of artificial oil bodies with or without interacting with EGCG (Figure 5). Consistent surface topology of artificial oil bodies alone or complexed with EGCG or rutin was observed, with much better resolution under a scanning electron microscope (Figure 6). Moreover, weak or strong aggregation was observed on artificial oil bodies alone or complexed with EGCG (Figure 6A,B). It seemed that the surface roughness of artificial oil bodies could be smoothed thoroughly after they were complexed with rutin (Figure 6C).

## 3. Discussion

Many tea competitions are regularly held in Taiwan to promote and market tea, as well as to motivate technical investigations for the improvement of tea quality [36]. Tea quality is firstly screened via sensory evaluation by committee members, and finally judged by a few (commonly 3–5) experts. The reliability of sensory evaluation for thousands of tea samples over several days is strongly challenged, and thus, there is an urgent need for a scientific means of assessing tea quality [37]. Astringency is an unfavorable sensation in tea, and it is regarded as a negative factor in tea quality evaluation [38,39]. Unfortunately, the astringent sensation is cumulative during tea consumption [40]; therefore, the accuracy of evaluating its astringency by continually tasting tea infusions is disputed. 

In this study, we estimated the relative astringency of eight tea catechins via an in vitro assay by measuring the thickness of the emulsion layer comprising artificial oil bodies complexed with each catechin. The results showed that gallate-type catechins were more astringent than non-gallate-type catechins, with an astringency order of ECG > EGCG > GCG > CG > EGC > EC > GC > C (Figure 1), which is in accordance with previous reported studies [10,14,41]. It seems that this in vitro assay for detecting relative astringency is practicable for assisting the sensory evaluation of tea infusions, although only one human salivary peptide (histatin 3) is employed to interact with catechins in the current assay system. Of course, this assay system can be further improved by including more salivary proteins and peptides on the surface of artificial oil bodies to mimic the saliva protein content proportionally.

Empirically, because of the increasing polyphenol content, the astringency of a tea infusion increases greatly when the brewing time is elongated [40]. Almost all tea lovers have experienced the strong astringency of a tea infusion when they forget their tea and it is left to brew. Surprisingly, the strong astringency of a tea infusion with an extended brewing time does not seem to be easily attenuated by dilution with water. To scientifically explore this experience, relative astringency was measured when a tea infusion was prepared from the same tea with brewing times of 10, 20, 30 and 40 min (Figure 2 and Figure 3). The results showed that an extension of the brewing time led to the increase in the catechin contents of a tea infusion, and thus elevated tea astringency. Interestingly, the enhanced proportion of gallate-type catechins was significantly higher than that of non-gallate-type catechins, indicating that tea astringency was elevated exponentially instead of proportionally, when brewing time was extended. It was likely that the release of non-gallate-type catechins (with relatively high water solubility) at the beginning of the brewing process might serve as detergent molecules to enhance the release of gallate-type catechins (with relatively low water solubility) from tea leaves. The enhanced release of gallate-type catechins in a tea infusion during the extension of brewing time provides a scientific explanation for why strong astringency is perceived when a tea infusion is prepared with an extended brewing time, and why it could not be effectively eliminated by dilution. It also rationalizes why tea lovers prefer to prepare a tea infusion in a small teapot, with a brewing time of less than a minute, and repeat the extraction of a tea infusion several times (e.g., 200 mL × 5), instead of simply setting tea leaves in a large bottle straightforwardly (1000 mL × 1).

According to sensory evaluation of the phenolic compounds of tea, catechins were assumed to trigger the sensation of puckering (rough) astringency, while flavonol glycosides were supposed to induce the sensation of velvety or silky (smooth) astringency [24,42,43,44]. In accordance with sensory evaluation, rough and smooth surface configurations were distinctly observed in this study using atomic force microscopy and scanning electron microscopy (Figure 4 and Figure 6) when EGCG (a major tea catechin) and rutin (a flavonol glycoside) were separately complexed with a major human salivary peptide, histatin 3, which was engineered on the surface of artificial oil bodies in this study. To the best of our knowledge, this is the first microscopy observation of the surface configuration of complex formed by a salivary peptide and a phenolic compound in a well-defined working system. Despite the fact that the experiment was representatively simplified by using one salivary peptide (histatin 3) and one phenolic compound (EGCG or rutin), this microscopy observation explained the distinct sensation of rough and smooth astringency for catechins and flavonol glycosides, respectively. It will be a challenging task to examine the surface configuration of complexes formed by saliva (or several major salivary peptides and proteins) and a mixture of tea phenolic compounds, particularly including both catechins and flavonol glycosides. Regarding the importance of beverage astringency in the sensation acceptance of consumers, it may be necessary for related industries to set up a platform of astringency evaluation [45]. Furthermore, the complicated configurations of different salivary peptides and proteins, such as mucin [46], should be further investigated, and the detailed interaction or surface configuration remains to be examined to gain a better understanding of tea astringency.

## 4. Materials and Methods

### 4.1. Chemicals and Materials

Catechin standards of EGC, EGCG, ECG, EC, GCG, GC, CG and C were purchased from Tokyo Chemistry Industry (Tokyo, Japan). High-performance liquid chromatography (HPLC)-grade acetonitrile and acetic acid glacial were purchased from Fisher Scientific (Fair Lawn, NJ, USA). Sesame oil was purchased from a local market. Chin-shin oolong tea was gifted by a local farmer, Mr. Jian-Cai Chang. Quercetin-3-O-rutinoside (rutin) was purchased from Sigma (St. Louis, MO, USA).

### 4.2. Generation of Artificial Oil Bodies for the In Vitro Assay of Tea Astringency

The cDNA fragment encoding a sesame caleosin lacking residues 101–115 was generated previously [47]. The histatin 3 DNA fragment was synthesized by Genewiz Inc. (South Plainfield, NJ, USA). The recombinant plasmid encoding the modified sesame caleosin fused with histatin 3 (approximately 29 kDa) engineered previously [35] was transformed to *Escherichia*
*coli* BL21 (DE3), and overexpression of the recombinant fusion protein was induced by adding 1 mM isopropyl β-D-thiogalactoside (IPTG). After IPTG induction (3 h), *E. coli* cells were lysed in 100 mM potassium phosphate buffer, pH 7.0, and separated into supernatant and pellet via centrifugation at 10,000× *g*. Proteins in the supernatant and pellet of *E. coli* cells were separately extracted, with the sample buffer containing 62.5 mM Tris-HCl, pH 6.8, 2% sodium dodecyl sulfate (SDS), 0.02% bromophenol blue, 10% glycerol and 5% β-mercaptoethanol, and resolved by polyacrylamide gel electrophoresis (PAGE) using 12% acrylamide (Figure 7). The fusion protein was insoluble and largely accumulated in a pellet of the lysed cell extracts. Proteins in the gel were visualized by staining with Coomassie blue R-250. The caleosin–histain 3 fusion protein eluted from the SDS-PAGE gel was quantitated by ImageJ 1.52a program [48]. This recombinant protein was collected and used in the artificial oil body production necessary for the astringency in vitro evaluation assay, according to the previous study [32]. The preparation used for the astringency assay was approximately 40 mg of artificial oil bodies in 1 mL of 100 mM potassium phosphate buffer with 50 mM KCl, pH7.0.

### 4.3. Preparation of Catechin Standards and Tea Infusion

Eight tea catechin standards of 2 mM were prepared in distilled water for the in vitro assay of astringency. Oolong tea leaves (3 g) were added to boiling water (150 mL) and kept in a water bath at 70 °C for 40 min. A tea sample of 1 mL was collected every 10 min. The four infusions prepared from the same oolong tea with brewing times of 10, 20, 30 and 40 min were subjected to an in vitro assay of astringency and HPLC analysis.

### 4.4. In Vitro Assay of Relative Astringency

Artificial oil bodies of 600 μL were mixed with an equal volume of catechin standard solutions (2 mM) or oolong tea infusions and then loaded into a 1.5 mL plastic cuvette, as described previously [32]. The flotation of aggregated artificial oil bodies forming a visible emulsion layer on the upper portion of the cuvette was observed after mixing for 15 min. Photos were taken with a digital camera after mixing for 45 min. Relative astringency was calculated according to the thickness of the visible emulsion layer measured by ImageJ 1.52a program [48].

### 4.5. Light Microscopy of Aggregated Artificial Oil Bodies

Aggregation of artificial oil bodies mixed with catechins or oolong tea infusions was observed under a light microscope (Eclipse E600, Nikon, Tokyo, Japan). The aggregated artificial oil bodies on the upper portion of the cuvette were sampled and kept at room temperature for 60 min prior to the light microscopy observation. The photos were taken and processed through ZEN 2.3 (blue edition, https://www.zeiss.com/microscopy/us/products/microscope-software/zen-lite.html) (accessed on 22 September 2019).

### 4.6. HPLC Analysis of Tea Infusion

Tea infusion was filtered through a 0.45 μm polyvinylidene difluoride membrane filter (PALL Corporation, Glen Cove, NY, USA). A high-performance liquid chromatography (HPLC) 600E system was equipped with 2996 Photodiode Array Detector, 600 Controller and 717 Autosampler (Waters Corporation, Milford, MA, USA). A Syncronis C18 column of 250 mm (length) × 4.6 mm (diameter) was used to analyze the tea infusion. The mobile phase consisted of 100% acetonitrile and distilled water with 0.5% acetic acid. The linear gradient started from 5% of 100% acetonitrile and 95% of 0.5% acetic acid. The gradient of the former switched to 15% at 30 min and 35% at 50 min, and ended with 5% at 60 min. The injection volume was 20 μL with flow rate at 1 mL/min. Ultraviolet (UV) absorbance of 280 nm was observed, as phenolic compounds were commonly seen at this wavelength.

### 4.7. Atomic Force Microscopy Imaging of Artificial Oil Bodies

To avoid interruption of imaging by precipitation or crystallization, artificial oil bodies were suspended in distilled water instead of the potassium phosphate buffer. Artificial oil bodies were mixed with 5 mM of EGCG and rutin, respectively. The mixture was dropped onto the polished side of a silicon wafer that was treated with oxygen plasma to ensure the surface was hydrophilic and the mixture spread uniformly. Samples were kept at 4 °C overnight for drying. The Bruker Dimension Icon atomic force microscopy system was used to observe the surface of artificial oil bodies and to construct the 3D topography. To obtain high-resolution images, tapping mode was selected to tip scan the surface with gentle force and less damage. The probe was adopted from a Nanoworld NCSTR series whose resonance frequency and spring constant were 160 kHz and 7.4 N/m, respectively. Optimal feedback control was used to enhance image quality. The scanning parameters were a scan rate of 0.3–0.6 Hz, a scan range from 5 to 20 μm and a resolution of 512 by 256 pixels. Images’ data were analyzed by the software NanoScope Analysis (Ver. 7.4, Filmetrics, Minneapolis, MN, USA), including the average of roughness (Ra), three-dimensional views and height information.

### 4.8. Scanning Electron Microscopy Imaging of Artificial Oil Bodies

Artificial oil bodies suspended in distilled water were mixed with 5 mM of EGCG or rutin. The mixture was dropped on the polished side of a silicon wafer that was treated with oxygen plasma and sputter coated with gold. The scanning electron microscopy system used in this study was Zeiss ULTRA plus with accelerating voltage of 3 kV. The working distance was 8.2 mm and magnification was 5.0 k.

### 4.9. Statistical Analysis

Data were shown as mean ± standard deviation (S.D.). One-way analysis of variance (ANOVA) and post hoc analysis of Tukey’s test were used to evaluate statistical significance. Statistical calculation was performed by SigmaStat (version 12.0, Systat Software, San Jose, CA, USA) with *p* < 0.05 (*), *p* < 0.01 (**) and *p* < 0.001 (***) defined to be statistically significant.

## Figures and Tables

**Figure 1 molecules-27-05679-f001:**
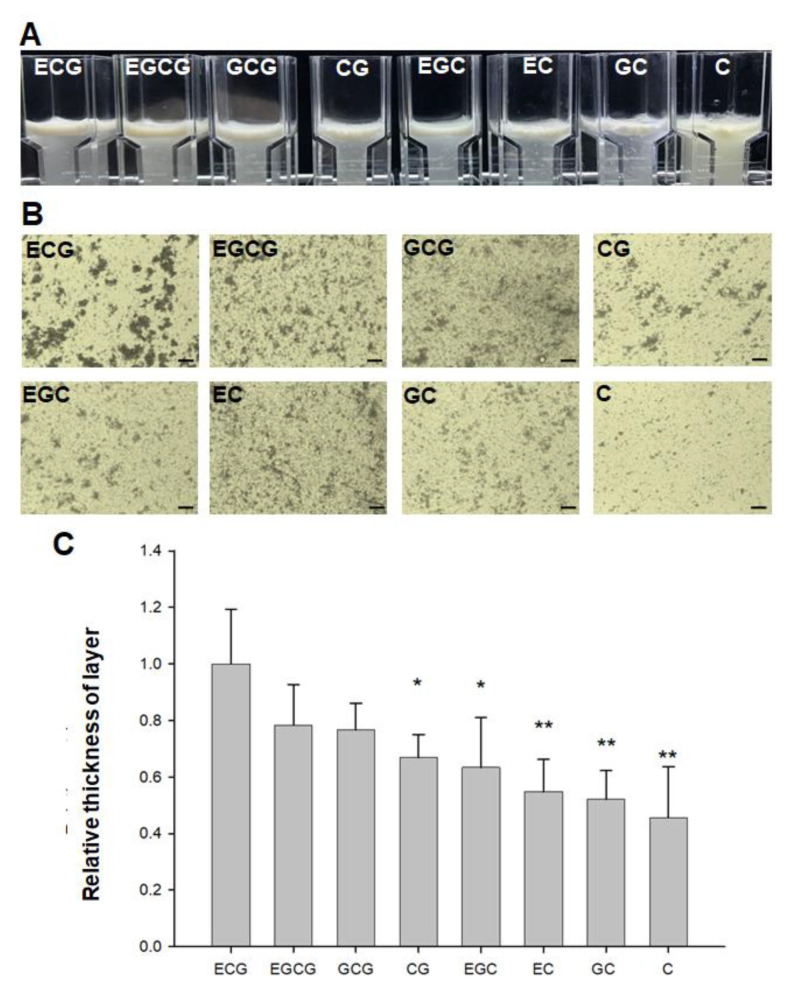
(**A**) Flotation of aggregated artificial oil bodies interacting with eight catechins in tea: ECG, EGCG, GCG, CG, EGC, EC, GC and C. Aggregated artificial oil bodies formed a visible emulsion layer on upper portion of the cuvette. (**B**) Light microscopy of aggregated artificial oil bodies interacting with eight catechins. Artificial oil bodies were allowed to interact with each catechin at room temperature for 60 min before taking the photos. Bars represent 40 μm. (**C**) The relative astringency of eight catechins. The relative astringency was calculated by measuring the thickness of emulsion layer in each catechin interaction, normalized with the thickness of emulsion in ECG interaction set as 100%. Statistical significance was processed by one-way ANOVA with Tukey’s test posttest (*n* = 3). * indicates *p* < 0.05, ** indicates *p* < 0.01.

**Figure 2 molecules-27-05679-f002:**
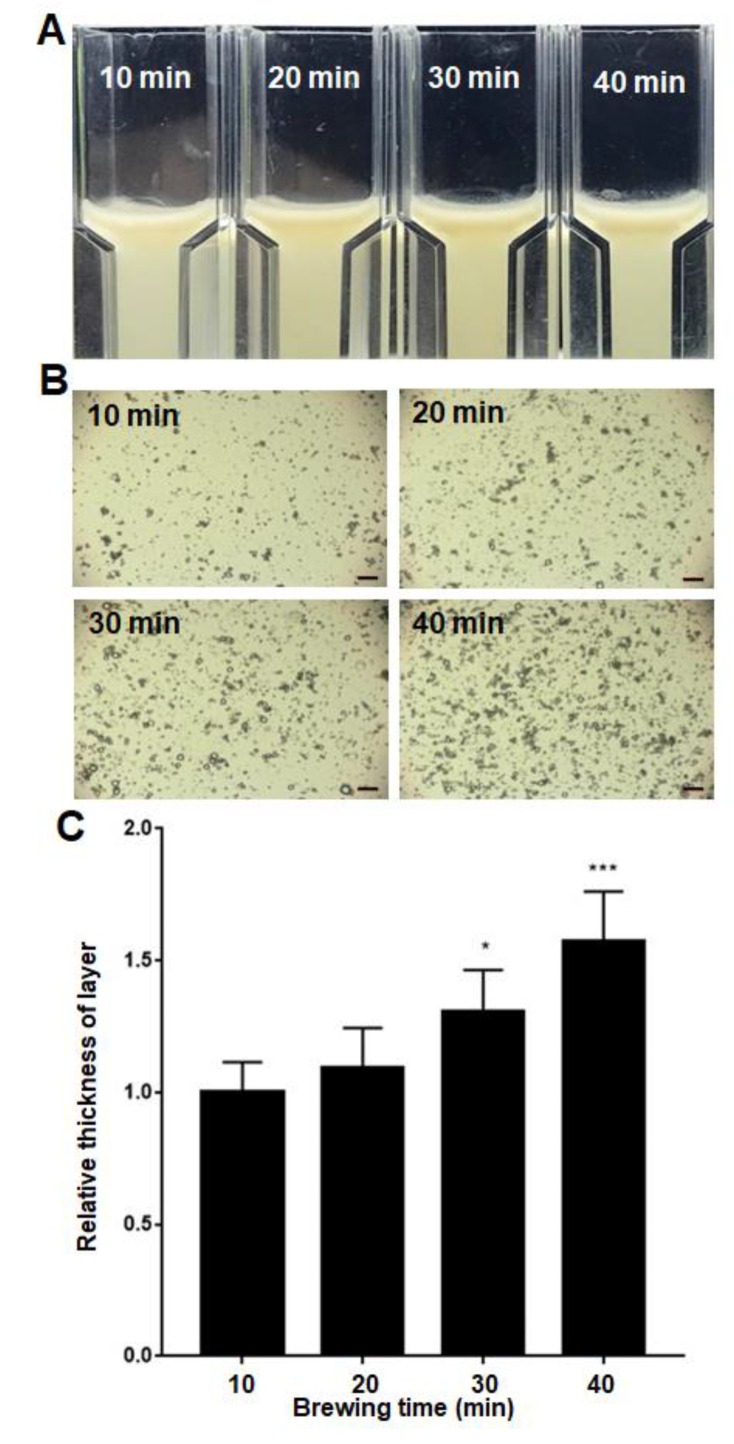
(**A**) Flotation of aggregated artificial oil bodies in tea infusion prepared with extended brewing time. A visible emulsion layer on the upper portion of the cuvette was observed for artificial oil bodies mixed with a tea infusion prepared via brewing for 10, 20, 30 or 40 min. (**B**) Light microscopy of artificial oil bodies in tea infusion. Aggregation of artificial oil bodies in the four tea infusions was observed at room temperature for 60 min before taking the photos. Bars represent 40 μm. (**C**) Relative astringency of the four tea infusions. The relative astringency was calculated by measuring the thickness of emulsion layer and normalized by setting the thickness of emulsion in 10 min-brewing tea infusion set as 100%. Statistical significance: * indicates *p* < 0.05, *** indicates *p* < 0.001 (*n* = 3).

**Figure 3 molecules-27-05679-f003:**
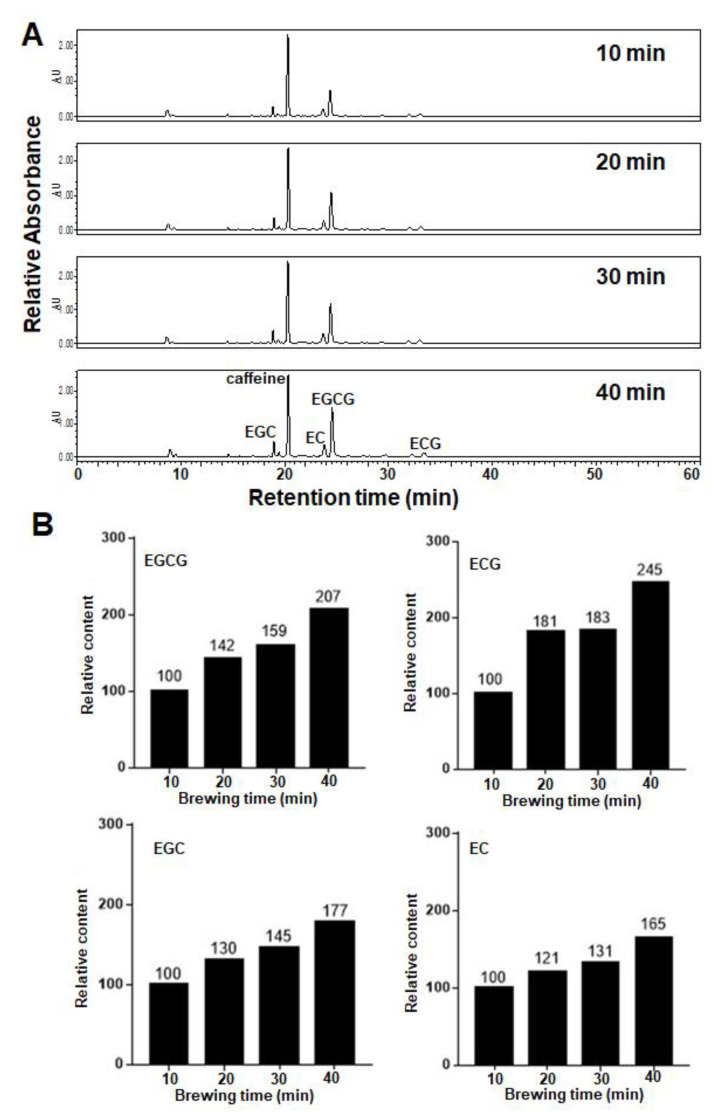
(**A**) HPLC profiles (OD 280 nm) of the four tea infusions prepared with different brewing time. The contents of four major catechins in infusion prepared by brewing the same tea for 10, 20, 30 or 40 min were analyzed and compared. The peaks of caffeine and the four major catechins, EGCG, EGC, ECG and EC, were indicated in the profile of tea infusion when they were brewed for 40 min. (**B**) Relative contents of EGCG, EGC, ECG or EC in the four tea infusions. The relative contents of EGCG, EGC, ECG and EC were estimated by calculating their peak areas in the HPLC profiles in (**A**). The content of each catechin in 10 min-brewing tea infusion was set as 100%.

**Figure 4 molecules-27-05679-f004:**
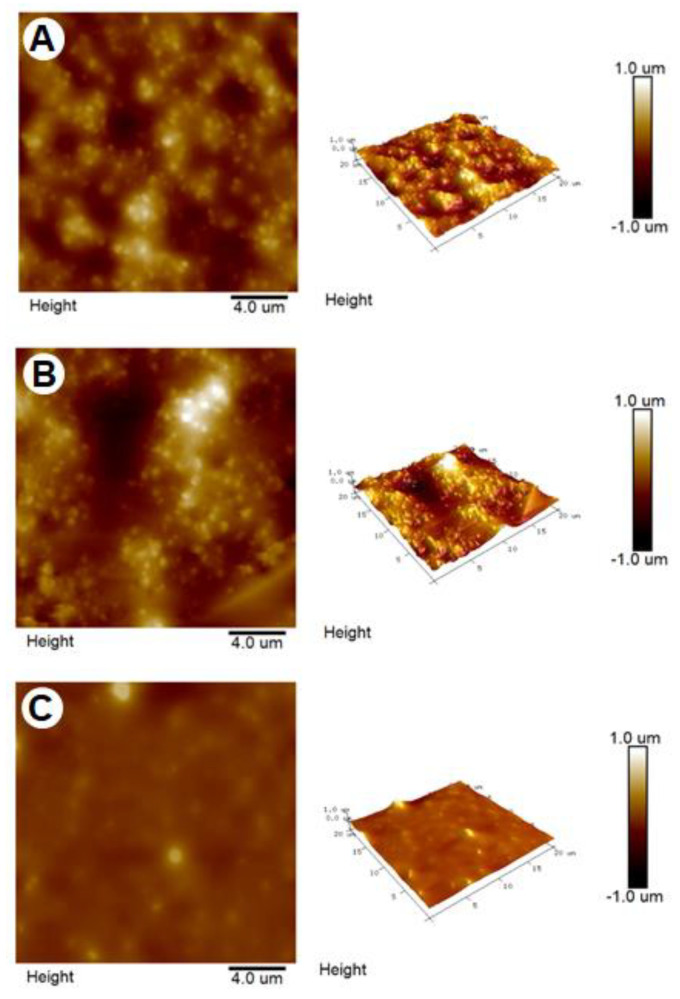
Topography images of artificial oil bodies alone (**A**) or complexed with EGCG (**B**) or rutin (**C**) in atomic force microscopy. Artificial oil bodies were mixed with or without 5 mM of EGCG or rutin and observed under the atomic force microscope.

**Figure 5 molecules-27-05679-f005:**
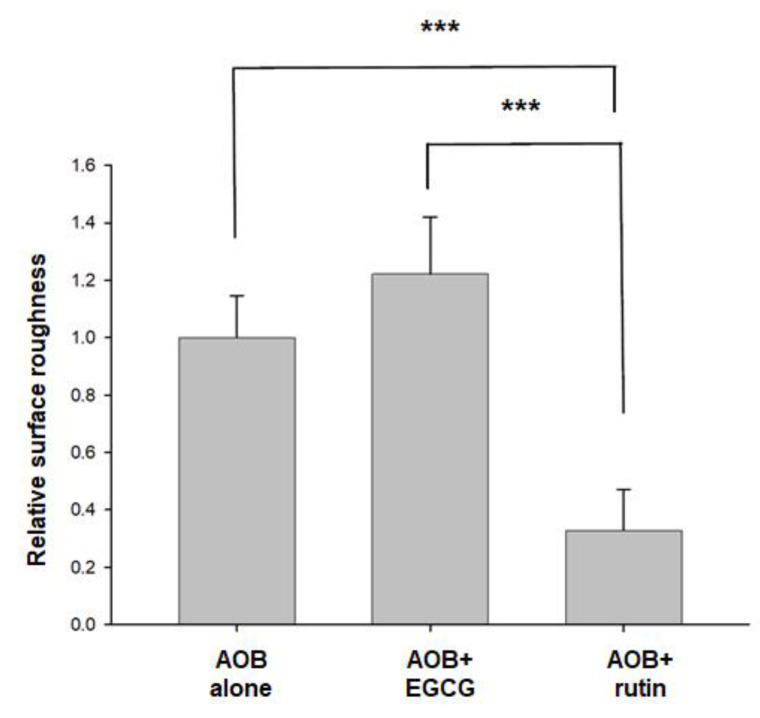
Relative surface roughness of artificial oil bodies (AOB) alone or complexed with EGCG or rutin. Surface roughness was analyzed by NanoScope Analysis (Ver. 7.4). Relative surface roughness was calculated by sampling (*n =* 5) images in Figure 5, and the average value of surface roughness of AOB alone was set as 100%. Statistical significance: *** indicates *p* < 0.001.

**Figure 6 molecules-27-05679-f006:**
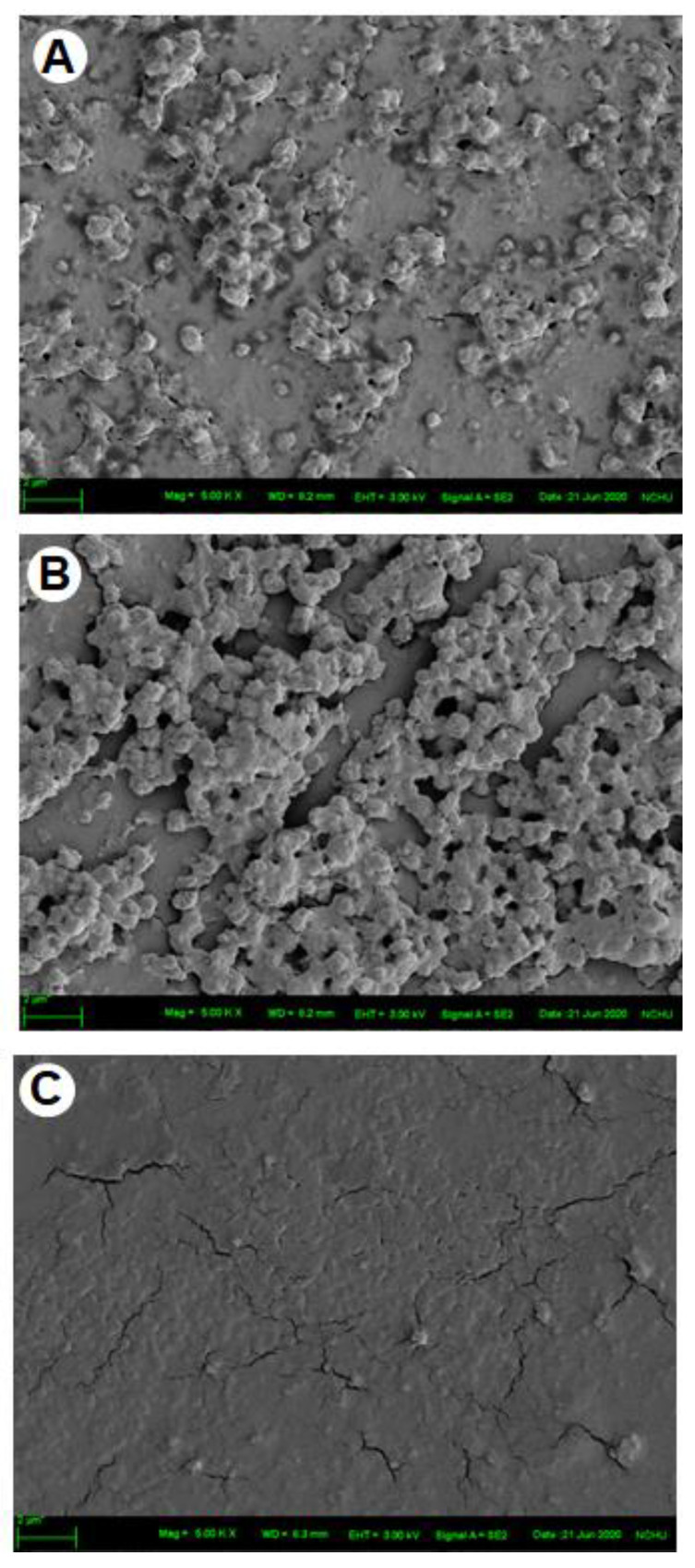
Topography images of artificial oil bodies alone (**A**) or complexed with EGCG (**B**) or rutin (**C**) in scanning electron microscopy. Artificial oil bodies were mixed with or without 5 mM of EGCG or rutin and observed under the scanning electron microscope (scale bar = 2 μm).

**Figure 7 molecules-27-05679-f007:**
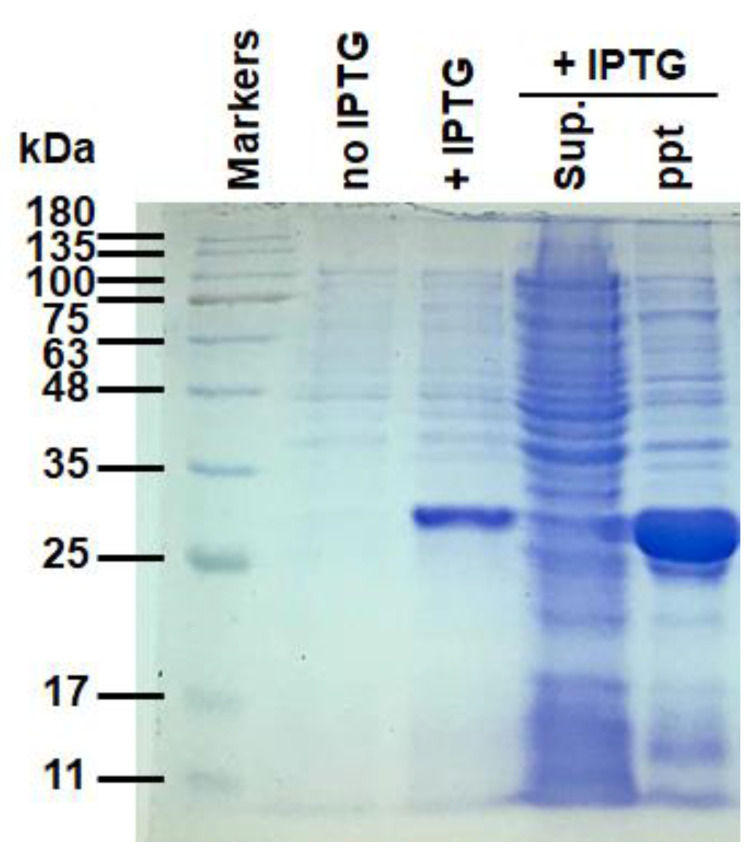
SDS-PAGE of the recombinant caleosin–histatin 3 overexpressed in *E. coli*. Total proteins of *E. coli* cells with caleosin–histatin 3 overexpressed after isopropyl β-D-1-thiogalactopyranoside (IPTG) induction were resolved in SDS-PAGE. Soluble (sup.) and insoluble (ppt) proteins extracted from *E. coli* cells were also analyzed. Labels on the left indicate the molecular masses of commercial marker proteins (Genemark, Taichung, Taiwan).

## Data Availability

Not applicable.

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
