# Peer review of "Exploring The Relative Astringency of Tea Catechins and Distinct Astringent Sensation of Catechins and Flavonol Glycosides via an In Vitro Assay Composed of Artificial Oil Bodies"

_molecules, 2022, doi:10.3390/molecules27175679_

Round 1

Reviewer 1 Report

In this study, the authors explored relative astringency of tea catechins and distinct astringent sensation of catechins and flavonol Glycosides via an in vitro assay composed of artificial oil bodies. The results suggested that catechins and flavonol glycosides induce the sensation of rough and smooth astringency in tea, respectively. The aim of the research is a good and is benefit for us to understand formation of the oral sensation of astringency of tea. The main comments are listed as below:

 1. The authors described in the introduction that “histatins and statherin are able to interact with phenolic compounds, leading to the oral sensation of astringency…”. How do these phenolic compounds interact with histatins and statherin? The action of mechanism is not clear. The authors should carry out more studies to reveal the mechanism.

2. Figure 4A. The x-axis title “Retenton time” should be “Retention time”.

3. Most of the references are cited in the Introduction section. More contents should be added to the Discussion section in comparison with previously reported studies.

Author Response

Please see the attached file, Answers to reviewer 1.

Reviewer 2 Report

The manuscript dials with an in vitro assay aimed to quantitatively evaluate the astringency capacity of beverages using artificial oil body droplets, covered by a recombinant surface protein, caleosin fused with histatin 3 (human salivary protein). The authors applied the assay to study the astringency of Oolong tea. The effect of the brewing time on the polyphenols content of tea and its astringency were evaluated by HPLC analysis and the proposed in vitro astringency assay. Furthermore, the author also studied the astringency power of the main polyphenols detectable in the Oolong tea (catechins derivates) by the proposed method and by the observation of the obtained aggregates through Scanning Electron, Atomic Force and Light Microscopy. Results are interesting even if part of them are already reported in a previous published paper (doi: 10.1016/j.jfda.2016.08.008) of the authors (correctly mentioned and reported as reference). On the all the scientific method is appropriate and I have only minor remarks which I listed below. I have a consideration about the feasibility of the proposed astringency in vitro assay, is it economically sustainable, considering that it is based on recombinant proteins and engineering artificial oil body?

Minor remarks

Line 72 In my opinion paragraph 2.1 should be moved in the material and methods section.

L79 Please change the phrase “This recombinant protein was harvested and used to generate artificial oil bodies that were used to serve as an in vitro assay system for the evaluation of tea astringency according to the previous study” as follow “ This recombinant protein was collected and used in the artificial oil body production necessary for the astringency in vitro evaluation assay, according to the previous study [32].

L157 correct typos “whem” instead of “when”

L195-198 “Unfortunately, astringency sensation ….of tea infusions” reformulate.

Author Response

Please see the attached file, Answers to reviewer 2.

Reviewer 3 Report

1. In the introduction, an overview of the current methods used to estimate astrigency level should be added.

2. In section 2.2, the DOE is flawed. Artificial oil bodies alone without any catechin must be included.

3. Have the authors compare your results of astrigency level with currently established ones such as human sensory methods? Without the comparision, there is not way to evaluate the results presented in this study.

4. In Fig 6, is there any significant differences between "AOB alone" and "AOB+EGCG". The authors can not neglect these results.

Author Response

Please see the attached file named Answers to reviewer 3.

Round 2

Reviewer 1 Report

The authors have addressed all my comments and revised their manuscript carefully. The manuscript can be accepted in its present form.

Author Response

Thanks for the positive recognition of our research work.